# At-Home Urea Breath Testing Demonstrates Increased Patient Uptake, High Satisfaction Rates, and Reduction in Carbon Emission Due to Eliminated Hospital Attendances, While Maintaining Diagnostic Accuracy for *H. pylori*

**DOI:** 10.3390/jcm14186598

**Published:** 2025-09-19

**Authors:** Conor Costigan, Edric Leung, Sandeep Sihag, Emmanuel Omallao, Deirdre McNamara

**Affiliations:** Department of Gastroenterology, Tallaght University Hospital, D24 NR0A Dublin, Ireland; edric.leung@tuh.ie (E.L.); mcnamad@tcd.ie (D.M.)

**Keywords:** telemedicine, near-patient testing, green endoscopy, urea breath test, *H. pylori*

## Abstract

**Background/Objectives**: Healthcare accounts for approximately 4.4% of global carbon emissions. Gastroenterology is a particularly heavy producer, with professional organisations outlining targets to move towards carbon neutrality. Missed hospital appointments, associated with poor medical outcomes, also represent physical and economic waste to the sector. COVID-19 expedited the shift toward virtual clinics, but tele-diagnostics have not expanded similarly. We aimed to assess the feasibility of a virtual C13 urea breath test clinic for *H. pylori* in Ireland. **Methods**: C13 urea breath test kits were provided to patients in the community, who were subsequently invited to book an online video appointment with a GI lab technician to assist them in performing the test at home. Completed tests were returned to the hospital via local GP, by post, or a specified hospital drop-off point, and analysed using our standard protocol. **Results**: 423 virtual appointments were reviewed. 135 (32%) were male, and the mean age was 42 years. The test positivity rate was 22%, similar to a matched in-person testing cohort (21%). In all, there were no non-attenders, and two cancellations. Virtual patients were more likely to attend their appointments (OR = 153.9, *p* = 0.0004) than in-person patients. Virtual UBT appointments saved 9943.5 Km of road journeys, equivalent to 254 person-hours of travel time and 1.24 metric tonnes of CO_2_. Additionally, 300 (71%) patients returned a feedback questionnaire, of which 276 (92%) rated the overall home breath test experience as ‘good’ or ‘excellent’. **Conclusions**: Home testing for *H. pylori* is effective, acceptable, and reduces both reliance on invasive procedures such as endoscopy and carbon emissions.

## 1. Introduction

Dyspepsia is one of the most common conditions presented to healthcare services in the general population, with some studies estimating a prevalence of up to 40%, with nearly half of those affected describing the symptoms as ‘moderate’ or ‘severe’ [1]. In 1999, the British National Health Service (NHS) estimated drugs for dyspepsia accounted for 10% of primary care drug expenditure [2].

Investigation of this common complaint is primarily by upper GI endoscopy (OGD); however, recently the demand for such services has increased in Ireland. In 2005, nearly 42,000 elective upper GI endoscopies were performed nationwide; however, by 2012, that number had risen to over 60,000 [3]. A recent cross-sectional study of 382,370 diagnostic upper GI endoscopies from the British Society of Gastroenterology suggested that 89.9% of procedures revealed only normal findings or minor pathology [4]. This suggests that the vast majority of OGDs are normal studies, highlighting the importance of appropriate patient selection to ensure the efficient use of healthcare resources. While non-invasive investigations of the upper GI tract have become commonplace and are designated as first-line investigations in many instances, their availability is highly dependent on location, and stool testing is often not acceptable to patients. For these reasons, diagnostic endoscopy services are often under pressure, with rising waiting lists, to deal with ever-increasing demand.

Healthcare accounts for approximately 4.4% of greenhouse gas emissions worldwide [5]. In Ireland, the national Health Service Executive (HSE) published its Climate Action Plan in 2023, aiming to be carbon neutral by 2050 [6]. Gastroenterology is a particularly heavy producer of emissions in the hospital setting. Evidence has suggested we are the third highest producer of hazardous waste in the hospital setting, and the second overall waste producers per clinical procedure, after radiology [7,8]. Endoscopy procedures alone accounted for an estimated carbon footprint equivalence of 85,768 tonnes of CO_2_ emissions due to their high water usage and single-use plastics [9].

More recently, with a spotlight on sustainability, there has been significant focus in the gastroenterology community on methods to reduce our carbon footprint [10,11]. Recent position papers from GI professional organisations across the world have outlined targets for the specialty, and its services, as we move towards carbon neutrality [12,13].

In addition, missed hospital appointments have been shown to lead to poor outcomes, including worsening health, delays to diagnosis and treatment, repeat presentations, and emergency healthcare usage, as well as a reduced quality of life [14]. Additionally, wasted space, personnel resources, and equipment adversely affect the healthcare system. In the US alone, missed clinical appointments cost millions of dollars annually [15,16]. Investigations into the causes for missed appointments have shown the problem is multifactorial; however, advances in telemedicine have the potential to overcome many of the most common, including lack of transportation or childcare, conflicts with work, or personal commitments [17,18]. While virtual clinics for initial preassessment, assessment, and follow up have become more common since the COVID 19 pandemic, with obvious benefits to both healthcare providers and to patients, the use of home diagnostics has not expanded at a similar rate. However, it may offer a multitude of benefits including improved patient access and uptake, as well as the obvious reductions in emissions and waste. In this study we aimed to develop and asses the feasibility of a virtual urea breath test clinic for *H. pylori* infection in an Irish population.

## 2. Materials and Methods

### 2.1. Study Design and Patient Selection

This prospective pilot study was designed to assess the feasibility of at-home UBT for the diagnosis of *H. pylori* infection. Primary outcomes included patient uptake/attendance at virtual appointment, *H. pylori* test positivity rate, and rates of sample analysis or other technical failure. Secondary outcomes included patient satisfaction with the at-home testing process and reduction in carbon emissions due to eliminated road transit from in-person hospital attendance.

The study was approved by the hospital quality improvement and audit committee (TUH/SJH JREC) as a quality improvement initiative, and so full ethical review was waived. All participants signed the standard hospital procedure consent form in order that their biological samples could be processed.

Patients were recruited by GPs in the community, where specific home UBT kits, which included 50 mg Urea (Diabact, Laboratoires Mayoly Spindler, Chatou, France), a paper straw, patient-specific labels, and four sample test tubes in a self contained box, were offered and provided directly to the patients from the local GP surgery and also from hospital booking systems by hospital staff, where at-home UBT kits were posted to the patients’ home addresses via the Irish national postal system “An Post”.

Inclusion criteria included any patient qualifying for UBT due to suspected *H. pylori* infection as well as the following: patients under 55 years of age with dyspepsia or acid reflux in the absence of ‘red flag’ or ‘alarm’ symptoms, patients >55 years of age or with alarm symptoms who declined or were unsuitable for upper GI endoscopy; and any patient previously treated for *H. pylori* infection to confirm successful eradication. Exclusion criteria included symptomatic patients with ‘alarm symptoms’ such as unexplained anaemia, weight loss, or dysphagia, active smokers, and those with a family history of gastric or oesophageal cancer.

### 2.2. At-Home Testing Process

In addition to at-home testing kits, patients also received a QR code and weblink to an online booking system, allowing them to make a booking, at a time of their own convenience, for a video call with a hospital-based GI lab technician or physiologist to guide and assist them throughout the process. Online information was given at the time of booking with regard to appropriate timeframes regarding stopping of PPIs (7 days) and antibiotics (28 days) in advance of performing the test.

At their chosen appointment times, patients received a video call from hospital staff, who discussed the test, access to results via the referring physician, and ensured appropriate abstinence from PPIs and antibiotics. Patients with recent PPI or antibiotic exposure were rescheduled in line with our in-person protocol.

Patients were guided through appropriately labelling their test tubes, performing the breath test pre- and post-ingestion of the provided urea tablet, in the comfort of their own homes or workplaces, as appropriate, and at their chosen time. Patients could choose the following return options for their completed test kits: by national postal service to the GI lab in Tallaght University Hospital, personal delivery to a predesignated 24 h drop-off point in the hospital, or returning their completed test to the local GP surgery for batch delivery to the hospital at a later date.

### 2.3. Analysis

Once returned, completed UBTs were processed by our standard protocol in batches of up to 40 patients, using an infrared analyser (HeliFANplus, Fischer ANalysen Instrumente GmbH, Leipzig, Germany). Testing was carried out by an appropriately trained GI physiologist or lab aide. A delta over baseline >4% was considered positive for *H. pylori* infection. Results from the test were returned to the referring physician electronically for recording and appropriate treatment, and a copy stored in the hospital’s electronic patient record.

Patient appointment information, basic demographics, and results were reviewed from the hospital patient management system, EPR, and GI lab database. Technical outcomes for the at-home testing group including test cancellation/non-attendance rates, overall *H. pylori* positivity rate, and rates of technical failure/inadequate sampling, were compared with a matched retrospective in-house cohort to ensure the technical validity of the at-home testing kits and process.

Statistical analysis using chi-square and odds ratio calculations were performed via SPSS v30 (IBM Corp, North Castle, Westchester County, NY, USA). Patient-reported outcomes and satisfaction with the home testing experience were also assessed by way of an optional questionnaire.

Additionally, distance from the hospital to the patients’ home, in kilometres, and transit time were calculated using the mapping software Google Maps v25 (Google Inc., Kirkland, WA, USA). Total CO_2_ emissions were calculated using an online calculator recommended by the Environmental Protection Agency (EPA) of Ireland [19]. Results in metric tonnes of CO_2_ were estimated using a reference petrol car of European average efficiency (6 L/100 km or 55 miles/gallon) [20].

## 3. Results

### 3.1. Demographics and At-Home Results

To date, records for 423 virtual attendees have been analysed. Of these, 135 (32%) were male. The mean age was 42 years (range 8–89 years, SD +/− 15.9 years). In total, in the home UBT cohort there were 0 non-attenders (0%), and 2 cancellations (0.5%). The overall positivity rate for *H. pylori* infection in the home cohort was 22% (n = 93). There were no cases of technical failure, or samples not returning a valid result.

This cohort was statistically similar to a matched retrospective cohort over the same time period (n = 691), representing 8 months of in-person testing, of which 242 (35%) were male, with a mean age of 44 years. Of these, a similar proportion, 145 (21%) were positive for *H. pylori* infection, and two technical failures were recorded, requiring tests to be repeated.

### 3.2. Attendence Rates

Over the whole duration of the at-home testing pilot, there were a total of 9096 in-person UBT appointments booked and offered to patients from our centre, of which only 5334 (59%) attended, while 2363 (26%) cancelled and subsequently rescheduled the appointment, and 1399 (15%) did not attend the appointment without giving notice. Patients booked for virtual appointments were significantly more likely to attend (OR = 153.9, 95% CI 9.6104–2467.5950, *p* = 0.0004), and less likely to cancel or reschedule (OR = 0.01, 95% CI 0.003–0.05, *p* < 0.001).

### 3.3. Travel Distances and CO_2_ Emissions

The total road distance saved by those attending virtual UBT appointments was 9943.5 Km, equivalent to 254 person-hours of travel time by car. The mean round-trip distance was 23.5 Km per patient. Overall, 1.24 metric tonnes of CO_2_ emissions were saved by performing home testing in the study cohort. In addition, 281 patients (90%), who were of working age (18–67 years) were saved taking half a day off work to attend an in-person appointment.

If data from the virtual UBT appointment experience were extrapolated out to the 9096 patients offered appointments during the data collection period, it would have saved 213,756 Km of road journeys, equivalent to 27.98 metric tonnes of CO_2_.

### 3.4. Patient Feedback

In total, 300 patients (71%) returned a completed feedback questionnaire. In all, 276 patients (92%) rated the overall home breath test experience as ‘good’ or ‘excellent’.

Furthermore, various aspects of the at-home testing process were individually assessed from the patients’ perspective. Respondents could describe each aspect of the process as ‘excellent’, ‘good’, ‘fair’ or ‘poor’. The following results were returned as outlined in Figure 1:(A)Patient information leaflet outlining the home UBT process; excellent or good: 289 (96%), fair or poor: 8 (3%), no response: 3 (1%)(B)The online booking system and self-selection of a virtual video appointment with GI laboratory technician; excellent or good: 255 (85%), fair or poor: 25 (8%), no response: 20 (7%)(C)UBT kit collection and drop-off process; excellent or good 286 (95%), fair or poor: 7 (2%), no response: 7 (2%)(D)Contents and ease of use of UBT testing kit; excellent or good: 291 (97%), fair or poor: 3 (1%), no response 6 (2%)(E)Video call with GI laboratory technician: excellent or good: 246 (82%), fair or poor: 34 (11%), no response 20 (7%)(F)Instructions; excellent or good: 292 (97%), fair or poor: 2 (1%), no response 6 (2%)

## 4. Discussion

Our pilot study confirms that virtual C13 UBT clinic appointments for *H. pylori* infection are feasible and acceptable to patients, increasing compliance and attendances while maintaining diagnostic accuracy. Furthermore, patient-reported outcomes were extremely positive overall, and analysis suggests significantly increased system efficiencies, particularly with respect to travel time saved for the patients, reduction in medical appointment related workplace absenteeism, and carbon emissions. All of these represent financial savings to the target patient population, the healthcare system, and the economy at large. We expect that, in the future, the virtual appointment with the GI laboratory staff may be automated by way of a prerecorded video instruction file available to patients undergoing testing, representing further savings in terms of staff efficiency.

The most significant results from this study, in addition to the above benefits, would be expanding access to a home version of this gold standard diagnostic test for *H. pylori* to areas currently not covered by a UBT service, thereby giving an alternative non-invasive diagnostic tool to local general practitioners and regional hospital physicians for their patients suffering with dyspepsia. Large scale roll out of this technology has the potential to greatly reduce pressures on local endoscopy services, which most often deal with the investigation of dyspepsia when UBT services are not available locally, and would represent a large a cost saving to the health service in terms of avoided unnecessary invasive procedures, potential complications, and hospital waste. Endoscopy service capacity could be more efficiently used in the investigation and management of other patients with other presentations or ‘red flag symptoms’. The full impact, however, and referring physician uptake of such a service in areas where C13 UBT is not already well established within diagnostic algorithms, remains to be seen. However, as most of the patients involved in this pilot study were local to our hospital, it is likely that time, resource and cost savings could be greater than those extrapolated above.

Greenhouse gas and hazardous waste emissions from healthcare are likely to continue to be significant due to the resource intensive nature of medical procedures and need for infection control, however the innovative use of technology can aid with patient selection, avoiding unnecessary procedures, and helping to offset some of these drawbacks as we move towards carbon neutrality. Similarly, efficiency in patient management systems, appropriate decentralisation, and minimising missed appointments allow healthcare systems to have more throughput and use staffing, equipment, and space resources more efficiently.

Limitations of this study include that it was a single-centre study, and due to its nature as a pilot study, only included small number of patients (n = 423). The study population involved were largely urban and local to our hospital, making returning completed test kits easy and timely. Although tests can be stored at room temperature for several days per manufacturers’ instructions, there are obvious quality control concerns regarding the stability of samples over time and with varying ambient temperatures. Previous studies have shown stability of collected samples at room temperature for timeframes of upward of 1 month [21,22].

Similarly, increasing distance from the end-analysis location adds to the transport time, and likelihood of physical damage to test tubes during transit. In the future we envision making use of the extensive courier network used to safely transport blood samples from community locations to hospital laboratories for testing, in order to avoid the possibility of damage in transit. A location in the nation’s capital city makes access to the high speed, reliable internet required to engage in the video call with the GI lab technician to perform the test at home more likely; this may not be available to those who live or work in rural areas. Similarly, 90% of participants were of working age and had a high degree of digital literacy, something which might make online booking and video calling more difficult in an older population.

Unfortunately, data was not collected on exact method of return as all completed UBT samples are stored in a specific storage unit in the GI lab to be run in batches of 40. CO_2_ calculations for this study are assuming all patients would have travelled from home, and via petrol car of average European efficiency. While many patients had their sample returned to the hospital by courier along with blood samples from their local GP practice or via national postal service, negating the need for travel at all, some will have deposited in their samples when attending another hospital in-person appointment, or visiting or accompanying another individual to the hospital or nearby appointment, effectively producing no additional emissions, and some patients travelled from their home, school, college or workplace before or after working hours via motor vehicles of all descriptions, bicycles, and public transport.

In Ireland, near-patient testing for routine laboratory diagnostics is already well established [23]. Tele-diagnostics and near-patient testing within the field of gastroenterology, however, is relatively new. Faecal Immunological Testing (FIT) has been at the vanguard of near-patient testing in gastroenterology and is now at the centre of most bowel screening (for colorectal cancer) and symptomatic patient vetting protocols for lower endoscopy procedures [24]. It has been shown to positively impact the diagnostic yield of colonoscopy in a safe manner, while reducing waiting lists for this resource-intensive procedure. The most recent uptake figures for those aged 59–69 in Ireland for the 2022–2023 timeframe stand at 46.4%, which may represent a cultural reluctance among our population to engage with outpatient stool testing [25]. In addition, data suggests that younger people—the target population for non-invasive *H. pylori* testing are less likely to engage with stool-based screening programmes, albeit for colorectal cancer screening [26]. As our acceptance rates for at-home UBT are significantly higher than those of many stool-based approaches, near 100%, this validates that the UBT approach is highly acceptable to Irish patients, while remaining the gold-standard test recommended by Irish and European *H. pylori* guidelines [27,28]

Similarly, at-home capsule endoscopy testing has been piloted in the UK with the 5G SUCCEEDS initiative and has been established in the US since the COVID-19 travel restrictions were introduced, with both programmes showing promising outcomes [29,30].

To our knowledge this is the first assessment of at-home breath testing, but both initiatives mentioned above are likely to be early examples in advance of a far wider adoption of tele- and virtual medicine across the healthcare sector globally. However, close attention needs to be paid to the green credentials of these initiatives, as not all near-patient or virtual healthcare programmes automatically imply reduced emissions or waste. For example, capsule endoscopy which relies than on more single use items for near patient testing than its in-hospital counterpart. For these, a hub and spoke model, where travel is reduced by proving tertiary services locally, may be more appropriate. Each diagnostic test and virtual model should be assessed on a case-by-case basis to quantify benefits for the health service, however our study would suggest that all forms of breath testing would be amenable to a tele-medicine format.

## 5. Conclusions

Our study has shown that near-patient breath testing is feasible, accurate, and acceptable to patients. It offers a model for further expansion of near-patient testing in gastroenterology. The impact of all diagnostics on emissions and waste should be at the forefront of our minds, including consideration of novel, near-patient testing where feasible.

## Figures and Tables

**Figure 1 jcm-14-06598-f001:**
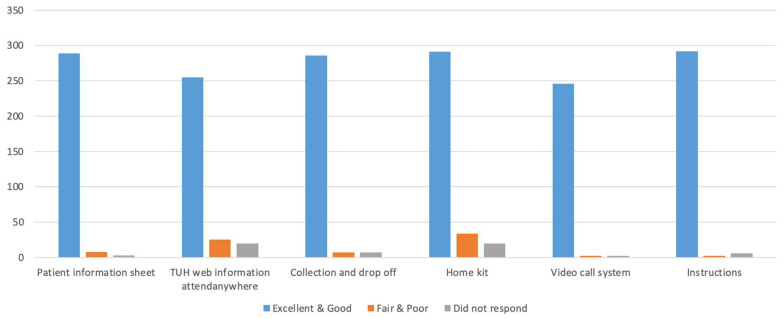
Patient Experience Questionnaire.

## Data Availability

The data presented in this study are available on request from the corresponding author due to Irish and European data protection legislation.

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
