# Peer review of "At-Home Urea Breath Testing Demonstrates Increased Patient Uptake, High Satisfaction Rates, and Reduction in Carbon Emission Due to Eliminated Hospital Attendances, While Maintaining Diagnostic Accuracy for H. pylori"

_jcm, 2025, doi:10.3390/jcm14186598_

Round 1
Reviewer 1 Report
Comments and Suggestions for Authors Dear EditorThis is an interesting to evaluate the usefulness of home UBT testing. The followings are my comments
#1. The idea of home testing is novel and interesting. Do you have any method to ensure the stability of the breath samples during the transfer process such as delay in mailing or temperature changes and thus impact the test accurracy?
#2. As testing for HP infection can be performed via stool antigen test, authors may provide some discussion on this point as iFOBT testing was discussed as near-patient testing for LGI colonoscopy in Ireland.
#3. How many patients require a video call with a hospital-based GI lab technician or physiologist to guide and assist them throughout the process in this study ? Do you compare the extra-time spend by the technician compared with conventional testing method ?
Author Response
Rear reviewer, thank you for taking the time to review and advise improvements to the submitted manuscript.
Comment 1: The idea of home testing is novel and interesting. Do you have any method to ensure the stability of the breath samples during the transfer process such as delay in mailing or temperature changes and thus impact the test accuracy?
Response 1: This is an interesting question, and likely to become more prominent if the service were to become exclusively a tele-medicine service and more distant patients were to be included and need to transport samples back to the GI lab for analysis.
The discussion (lines 299-303) of the manuscript has been updated to include "Although tests can be stored at room temperature for several days per manufacturers’ instructions, there are obvious quality control concerns regarding the stability of samples over time and with varying ambient temperatures. Previous studies have shown stability of collected samples at room temperature for timeframes of upward of 1 month [21,22]... In the future we envision making use of the extensive courier network used to safely transport blood samples from community locations to hospital laboratories for testing to avoid the possibility of damage in transit'.
Comment 2: As testing for HP infection can be performed via stool antigen test, authors may provide some discussion on this point as iFOBT testing was discussed as near-patient testing for LGI colonoscopy in Ireland.
Response 2: Testing via stool antigen for HP is also validated, although UBT remains the preferred testing, particularly for post-eradication testing. In Ireland we have recently rolled out "BowelScreen" a national stool-based screening programme for colorectal cancer, as well as triage systems for those referred for colonoscopy. Unfortunately uptake of these tests remains low, with some data showing younger people are less likely to engage, and UBT remains the preferred test for HP in most guidelines. The discussion is updated to include this discussion (line 332-340)
"The most recent uptake figures for those aged 59-69 in Ireland for the 2022-23 timeframe stand at 46.4%, which may represent a cultural reluctance among our population to engage with outpatient stool testing[25]. In addition, data suggests that younger people – the target population for non-invasive H. pylori testing -are less likely to engage with stool-based screening programs, albeit for colorectal cancer screening [26]. As our acceptance rates for at-home UBT are significantly higher than those of many stool-based approaches, near 100%, this validates that the UBT approach is highly acceptable to Irish patients, while remaining the gold-standard test recommended by Irish and European H. pylori guidelines [27, 28]"
Comment 3: How many patients require a video call with a hospital-based GI lab technician or physiologist to guide and assist them throughout the process in this study ? Do you compare the extra-time spend by the technician compared with conventional testing method ?
Response 3: In this study all patients received a virtual call from hospital staff to guide them through the testing process, and so the time for testing itself is the same as if the patient attended in-person testing. In the future we envisage a prerecorded video that patients can access, freeing up the GI technician and representing further efficiency savings tot he health system.
This is updated to reflect that statement "We expect that, in the future, the virtual appointment with the GI laboratory staff may be automated by way of a prerecorded video instruction file available to patients undergoing testing, representing further savings in terms of staff efficiency"
Line 263-266
Reviewer 2 Report
Comments and Suggestions for Authors
Interesting study, but I am concerned what happened to the patients who were positive for the UBT. Did they visit GP clinic or hospital, take gastroscopy and have eradication treatment? Or did they just receive the eradication medication by mail?
Saving unnecessary endoscopy is an important issue, but at least UBT positive patients should undergo gastroscopy to check gastric cancer and peptic ulcer.
I would like to know the reason why the authors chose UBT, instead of urine antibody or stool antigen as H. pylori test at home.
“Virtual UBT” in the title is misleading, because it suggests that the participants did not have real UBT. More simply, “UBT at home” seems better.
Author Response
Dear reviewer 2, thank you for your review of our study and advice to improve the manuscript prior to publication.
Comment 1: I am concerned what happened to the patients who were positive for the UBT. Did they visit GP clinic or hospital, take gastroscopy and have eradication treatment? Or did they just receive the eradication medication by mail?
Response 1: This study only deals with the technicalities of performing UBT testing. For the purposes of this study, similar to blood tests or histological analysis, test results were returned to the referring physician for appropriate treatment - either in primary general practice, physicians in local secondary care, or our own tertiary centre. The methods section (line 179/180) describes "Results from the test were returned to the referring physician electronically for recording and appropriate treatment, and a copy stored in the hospital’s electronic patient record." lab technicians cannot assess medical histories, antibiotic allergies or prescribe such medications.
Comment 2: Saving unnecessary endoscopy is an important issue, but at least UBT positive patients should undergo gastroscopy to check gastric cancer and peptic ulcer.
Response 2: Current guidelines have for many years recommend against routine endoscopic testing of HP+ patients unless persistent symptoms despite treatment, later age of onset or presence of 'alarm' features. This is the hallmark of the test-and-treat approach adopted in the late 90s and early 2000s in Europe and North America.
"It is widely accepted that endoscopy should be reserved for patients with symptom onset after 50 (45–55)years of age, those who have alarm features and all patients who fail empirical antisecretory therapy or test-and-treat strategy fails" - Maastricht VI guidelines, Gut, 2022.
Similar recommendations exist in Irish, European, British, American, and Canadian guidelines. However, in some higher risk populations, such as in some asian countries like Japan and Korea have different thresholds for endoscopy.
Unfortunately in Ireland, UBT is often unavailable locally, and stool testing is often poorly uptaken and unacceptable to patients, and so those with dyspepsia are referred to endoscopy, which guidelines suggest should not be performed. Stopping these endoscopies is one of the aims of the study- to improve compliance with guidelines
Comment 3: I would like to know the reason why the authors chose UBT, instead of urine antibody or stool antigen as H. pylori test at home.
Response 3: While some experimental urinary testing for HP exists, none have been adopted in widespread practice. UBT and stool-antigen are however widely accepted non-invasive tests. UBT is widely accepted as the gold standard non-invasive test for this procedure, and 'The second Irish Helicobacter pylori Working Group consensus' statement 7 recommends "Post-eradication treatment testing must be performed. If gastroscopy is not required, a UBT is recommended for post-eradication treatment testing. If the UBT is unavailable, the monoclonal stool antigen test is an alternative if locally validated."
This is due to concern regard the reliability of different monoclonal and polyclonal antigen testing kits are used, as these are not standardised, and whether rapid immunochromatography or enzyme immuno-assays are used, and the theory that post-eradication testing with stool may give false positives due to the shedding of antigens from dead bacteria, whereas UBT assess live bacterial metabolism.
Comment 4: “Virtual UBT” in the title is misleading, because it suggests that the participants did not have real UBT. More simply, “UBT at home” seems better.
Response 4: On reflection, the authors agree with this statement. The breath test itself is not virtual, only the appointment.
'Virtual' has been removed from the title.
Reviewer 3 Report
Comments and Suggestions for Authors
In the present study Costigan et al explored the possibility of urea breath test performed at home for H. pylori diagnosis. They showed that this strategy dramatically reduced the risk of appointment missing and lowered ambiental impact in terms of CO2 produced. Main comments:
1) The Background paragraph of the Abstract is too long.
2) Page 3 lines 128-130: more details about the historical cohort should be provided, also in Methods section regarding patients selection.
3) Statistics was not described in Methods section (for instance, OR was calculated).
4) Paragraph 3.2: I did not understand whether the CO2 saved was calculated by subtracting the CO2 that was consumed by those who carried the samples to central hospital (net CO2). If not, this calculation is necessary.
5) How were pre-test recommendations (e.g. stop taking PPI) given to patients?
6) Please report mean time from sample collection to lab analysis.
Author Response
Dear reviewer 3: Thank you for your advice regarding improvement to the manuscript.
Comment 1: The Background paragraph of the Abstract is too long.
Response 1: We agree with this statement. A more succinct introduction has been provided. It has been reduced from 9 lines to 6 lines.
Comment 2: Page 3 lines 128-130: more details about the historical cohort should be provided, also in Methods section regarding patients selection.
Response 2: This is an oversight on behalf of the authors. Inclusion and exclusion criteria have been added to the methods section (line 150-156)
Further information is also given for the retrospective cohort (lines 208-211)
Larger numbers were not assessed for this comparison due to time constraints in downloading each individual PDF file containing results in a retrospective manner from the hospital EPR. These results are also consistent with a service evaluation of 8 years worth of data from our centre showing annually that 38-45% of attendees are male with annual positivity rates varying from 18-24% over the past 8 years in our centre.
Comment 3: Statistics was not described in Methods section (for instance, OR was calculated).
Response 3: Thank you for pointing this out. Methods now includes lines 185/6:
"Statistical analysis using chi-square and odds-ratio calculations were performed via SPSS (IBM Corp, North Castle, NY, USA)."
Comment 4: Paragraph 3.2: I did not understand whether the CO2 saved was calculated by subtracting the CO2 that was consumed by those who carried the samples to central hospital (net CO2). If not, this calculation is necessary.
Response 4: Unfortunately, data was not collected on exact method of return as all completed UBT samples are stored in a specific storage unit in the GI lab to be run in batches of 40. CO2 calculations for this study are assuming all patients would have travelled from home, and via petrol car of average European efficiency. While many patients had their sample returned to the hospital by courier along with blood samples from their local GP practice or via national postal service, negating the need for travel at all, some will have deposited in their samples when attending another hospital in-person appointment, or visiting or accompanying another individual to the hospital or nearby appointment -effectively producing no additional emissions, and some patients travelled from their home, school, college or workplace before or after working hours via motor-vehicles of all descriptions, bicycles and public transport.
This has been updated in criticisms of the study under discussion line 314-324.
Comment 5: How were pre-test recommendations (e.g. stop taking PPI) given to patients?
Response 5: Verbal information was given to patients at the time of initially discussing the test with their referring GP / physician, a written PIL was provided with the at-home test kit and available on the hospital website at any time. Additionally, these medications were assessed as part of a virtual PIL during accessing the online booking system and patient rescheduled if recent exposure to PPI or antibiotics - in line with out in-person testing protocol.
This is updates in the text line 161-167
Comment 6: Please report mean time from sample collection to lab analysis.
Response 6: Mean time to testing was not recorded during data collection for this study, as return plans were discussed with between the GI lab staff and the patient during the vitrual consultation. However the body of the text is updated to reflect data that studies have shown stability of collected samples at room temperature for timeframes of upwards of 1 month [21,22]. - Line 300-303
Round 2
Reviewer 3 Report
Comments and Suggestions for Authors
Answers were fine